# Solid Polymer Electrolytes with Flexible Framework of SiO_2_ Nanofibers for Highly Safe Solid Lithium Batteries

**DOI:** 10.3390/polym12061324

**Published:** 2020-06-10

**Authors:** Jin Cui, Zehao Zhou, Mengyang Jia, Xin Chen, Chuan Shi, Ning Zhao, Xiangxin Guo

**Affiliations:** College of Physics, Qingdao University, Qingdao 266071, China; 18438610159@163.com (J.C.); zehaozhou@163.com (Z.Z.); mengyangjia2020@163.com (M.J.); chenxin6626@126.com (X.C.); N.zhao@qdu.edu.cn (N.Z.)

**Keywords:** solid polymer electrolytes, polyethylene oxides, three-dimensional SiO_2_ nanofibers, safety, solid lithium batteries

## Abstract

Composite electrolytes consisting of polymers and three-dimensional (3D) fillers are considered to be promising electrolytes for solid lithium batteries owing to their virtues of continuous lithium-ion pathways and good mechanical properties. In the present study, an electrolyte with polyethylene oxide–lithium (bis trifluoromethyl) sulfate–succinonitrile (PLS) and frameworks of three-dimensional SiO_2_ nanofibers (3D SiO_2_ NFs) was prepared. Taking advantage of the highly conductive interfaces between 3D SiO_2_ NFs and PLS, the total conductivity of the electrolyte at 30 °C was approximately 9.32 × 10^−5^ S cm^−1^. With a thickness of 27 μm and a tensile strength of 7.4 MPa, the electrolyte achieved an area specific resistance of 29.0 Ω cm^2^. Moreover, such a 3D configuration could homogenize the electrical field, which was beneficial for suppressing dendrite growth. Consequently, Li/LiFePO_4_ cells assembled with PLS and 3D SiO_2_ NFs (PLS/3D SiO_2_ NFs), which delivered an original specific capacity of 167.9 mAh g^−1^, only suffered 3.28% capacity degradation after 100 cycles. In particular, these cells automatically shut down when PLS was decomposed above 400 °C, and the electrodes were separated by the solid framework of 3D SiO_2_ NFs. Therefore, the solid lithium batteries based on composite electrolytes reported here offer high safety at elevated temperatures.

## 1. Introduction

Electric vehicles (EVs) are regarded as one of the promising transportations to reduce the consumption of fossil energy [1,2,3]. However, EVs suffer mileage and safety issues due to the capacity limits and security threats of liquid-electrolyte-based lithium-ion batteries (LIBs) [4,5,6,7,8,9,10]. Solid lithium batteries (SLBs) show great potential in breaking through the safety and capacity limitations of traditional LIBs [11,12,13,14,15] as they enable the use of high-voltage cathodes and Li metal anodes with the help of solid-state electrolytes (SSEs) [16]. Among numerous kinds of SSEs, polyethylene oxide (PEO)-based electrolytes have received extensive attention owing to their usable lithium-ion conductivities at elevated temperatures and acceptable stability against lithium anodes [17,18,19,20,21,22,23]. However, they also have shortcomings, such as an oxidization potential smaller than 4.0 V and low mechanical strength [24].

It has been demonstrated that the dispersion of ceramic nanoparticles or nanofibers, such as SiO_2_, Al_2_O_3_, Li_6.4_La_3_Zr_1.4_Ta_0.6_O_12_ (LLZTO), or Li_1.5_Al_0.5_Ge_1.5_(PO4)_3_ (LAGP), into the polymer matrices can increase the ionic conductivities, broaden the electrochemical windows, and improve the mechanical strengths of PEO-based solid polymer electrolytes (SPEs) [24,25,26,27,28]. In addition, it has also been shown that the interaction between PEO and fillers are dependent not only on different filler materials but also the morphology and size of fillers. However, one of the serious problems of using such granular fillers is that the brittle and flammable PEO is easily decomposed by thermal shock, and the separated fillers may lead to the thermal runaway of SLBs. Moreover, these granular fillers tend to agglomerate, resulting in the inhomogeneous electrical field influencing lithium-ion migration. This is a key factor in the growth of lithium dendrites [29,30,31].

To overcome the aforementioned shortcomings, three-dimensional (3D) active fillers, such as Li_6.4_La_3_Zr_2_Al_0.2_O_12_ (Al-LLZO) and Li_0.35_La_0.55_TiO_3_ (LLTO), have been introduced into SPEs [32,33,34,35]. However, the 3D frameworks of active fillers are usually fragile; they have to be hundreds of microns in thickness in order to be freestanding. Moreover, the processes of preparing these active fillers are expensive and complex. These shortcomings therefore limit their application. In the present study, we designed and prepared three-dimensional SiO_2_ nanofibers (3D SiO_2_ NFs) as fillers instead of the fragile active fillers using the electrospinning method. The 3D SiO_2_ NFs could maintain excellent flexibility and mechanical strength with thickness as small as 25 μm. In combination with PEO–lithium (bis trifluoromethyl) sulfate–succinonitrile (PLS), which is Li^+^ conductive [36,37], films of composite electrolytes with thickness of 27 μm were fabricated by the casting method. These composite electrolytes showed good conductivity, ability to suppress dendrite growth, and high safety at elevated temperatures. 

## 2. Experiments

### 2.1. Materials

Tetraethyl orthosilicate (TEOS), polyvinyl alcohol (PVA, *M*_w_ = 66,000), polyvinylidene fluoride (PVDF), phosphoric acid (H_3_PO_4_), and polyethylene oxide (PEO, *M*_w_ = 600,000) were purchased from Aladdin Reagent (Shanghai, China). Lithium (bis trifluoromethyl) sulfate (LiTFSI) (99.95%), anhydrous acetonitrile (ACN), and succinonitrile (SCN, 99%) were bought from Sigma-Aldrich (Tianjin, China). All materials were used without further purification.

### 2.2. Preparation of the Composite Electrolytes

For fabrication of SiO_2_ particles and 3D SiO_2_ NFs, refer to our previous works [38,39]. PEO/LTFSI/SCN with a mass ratio of 2:1:0.4 were added into the ACN (80% in mass ratio) solution and continuously stirred for 12 h. Then, the homogenized solution was cast into the 3D SiO_2_ NF membrane with a scraper for the preparation of PLS/3D SiO_2_ NFs or directly coated onto the surface of a Teflon plate for the PLS fabrication. The same mass ratio (equal to the 3D SiO_2_ NFs) of SiO_2_ particles were added to the above solution and coated onto the Teflon plate surface for the preparation of PLS/SiO_2_ particles. All the electrolytes were stored in a vacuum oven at 60 °C for 24 h to evaporate the solvent (the mass of the electrolyte was approximately 4.0 mg cm^−2^) and then used for the tests. 

### 2.3. Electrode Preparation and Cell Assembly

The battery performance was tested using coin cells. LiFePO_4_ (LFP)/Super-P/PLS/PVDF were mixed together with a mass ratio of 70:10:10:10 for the cathode preparation. The loading of LFP was 1.0 mg/cm^2^. Lithium metal was used as the anode and as-prepared membranes were used as the solid electrolytes. The same electrodes were used for preparing the punch cells. The cells were treated at 60 °C for 10 h to further improve the contact.

### 2.4. Characterization of the Electrolytes

The surface and cross-sectional morphologies of the PLS, PLS/SiO_2_ particles, and PLS/3D SiO_2_ NFs were investigated with a field-emission scanning electron microscope (FE-SU4800, Hitachi, Tokyo, Japan). The mechanical strength of SPEs was measured with a tensile machine (Instron-5500R, Instron Corporation, Princeton, NJ, USA). The AC impedance of PLS, PLS/SiO_2_ particles, and PLS/3D SiO_2_ NFs was measured by sandwiching the electrolytes between two stainless electrodes with an electrochemical workstation (Solartron, SI-1260, Shanghai, China) over a frequency range of 100 kHz to 1 Hz. The ionic conductivity of electrolytes was calculated by Equation (1): (1)θ=L/RS
where *R*, *L*, and *S* are the resistance, thickness, and area of the electrolytes, respectively. The area specific resistance (ASR) was calculated by Equation (2):(2)A=L/θ
where *L* and θ are the thickness and ionic conductivity of the electrolytes, respectively. Thermogravimetric analysis (TGA) and differential scanning calorimetry (DSC) of polymers and electrolytes were taken by a thermogravimetric analyzer (TA, SDT-650, New Castle, DE, USA) with a heating rate of 10 °C min^−1^. The electrolytes were put into a muffle furnace with a heating rate of 10 °C min^−1^ during measurement of AC impedance for the shutdown testing. Electrochemical stability of the electrolytes was studied using the linear sweep voltammetry (LSV) method with a range of 0–6 V at 0.5 mV s^−1^. Each electrolyte film was clamped between a stainless-steel electrode and a lithium metal counter electrode. Cells with PLS/SiO_2_ particles and PLS/3D SiO_2_ NFs were prepared to investigate the cycling and rate performances with the battery test equipment (LAND-V34, Land Electronic, Wuhan, China). The Li | PLS/3D SiO_2_ NFs | LFePO_4_ punch cells were charged to 3.7 V at 60 °C. Afterward, they were treated from 60 to 200 °C in an oven with a heating rate of 10 °C·min^−1^ to monitor the open-circuit voltage (OCV). Subsequently, Li | PLS/3D SiO_2_ NFs | LFePO_4_ were recharged and discharged at 60 °C.

## 3. Results and Discussion

Scanning electron micrographs of the prepared PLS/SiO_2_ particles and PLS/3D SiO_2_ NFs are shown in Figure 1 and Appendix A. An agglomeration of SiO_2_ nanoparticles in the PLS can be seen in Figure 1a–c. Due to the uneven distribution of mechanical stress and electric field, inevitable clustering damaged the performance of the electrolyte [19]. On the contrary, SiO_2_ NFs were evenly distributed in the main body of the PLS (Figure 1d–e), which can be attributed to the rigid structure of 3D SiO_2_ NFs. It should be noted that the minimum thickness of freestanding PLS/SiO_2_ particle membrane prepared by the flow coating method is 50 μm [40]. In contrast, in the case of 3D SiO_2_ NFs, the thickness of PLS/3D SiO_2_ NFs can be controlled below 27 μm. The application of thinner SPEs in SLBs is vital to achieve improved energy density. 

The stress–strain curves of the 3D SiO_2_ NFs and SPEs are illustrated in Figure 2. A mechanical strength of 0.6 MPa was observed in the PLS. The poor mechanical performance is one of the most critical obstacles for the practical application of PEO-based SPEs, such as PLS. Under the framework of 3D SiO_2_ NFs with a mechanical strength of 3.7 MPa, the maximum stress of PLS/3D SiO_2_ NFs reached 7.4 MPa. PLS/3D SiO_2_ NFs with enhanced mechanical properties are favorable for safety and show great potential in meeting the requirements of battery assembly. However, compared to the original PLS, the mechanical strength of PLS/SiO_2_ particles was even lower (0.4 MPa). The PLS/SiO_2_ particles were prone to break at the point of absence of PEO matrix due to the agglomeration of SiO_2_ particles (Figure 1c).

The AC impedances of PLS, PLS/SiO_2_ particles, and PLS/3D SiO_2_ NFs are demonstrated in Figure 3a. The ionic conductivity of PLS, PLS/SiO_2_ particles, and PLS/3D SiO_2_ NFs at 30 °C were 4.72 × 10^−5^, 5.28 × 10^−5^, and 9.32 × 10^−5^ S cm^−1^, respectively. With the thickness of 50, 50, and 27 μm, the corresponding ASR were 105.9, 94.7, and 29.0 Ω cm^2^, respectively. It has been demonstrated that the ionic conductivities of SPEs are interrelated with the percolation behavior. This is due to adsorption on the inorganic–organic interface, leading to an increase in the concentration of mobile carriers [25,41]. Therefore, Li^+^ can move along the interfaces between SiO_2_ fillers and the PEO chains as well as pass through the polymer segments. The results indicate that the cross-linked interfaces between PEO and 3D SiO_2_ NFs facilitate a higher ionic conductivity than the isolated and agglomerated interfaces formed between PEO and SiO_2_ particles. Besides, PLS/3D SiO_2_ NFs with a low ASR value is conducive to the improvement of battery performance [11]. In addition to a favorable ionic conduction, considerable electrochemical stability of electrolyte is needed to satisfy the requirements of SLBs. LSV was carried out, and the result is shown in Figure 3b. An obvious oxidation process was observed at 4.35 V in the PLS, which was the reaction of PEO decomposition. After the introduction of SiO_2_ particles, the initial oxidation potential of PLS/SiO_2_ particles could be extended to 4.50 V. The SiO_2_ particles were sintering under 600 °C for 2 h, which contained negligible amount of moisture. During the preparation of composite electrolytes, the environment was kept dry in the drying room. In this case, the SiO_2_ would tend to absorb water in the organic component. The electrochemical stability enhancement might be due to the removal of trace amounts of moisture and acidic impurity in the electrolyte with the SiO_2_. Regarding PLS/3D SiO_2_, an enhanced electrochemical stability was achieved, showing an oxidation current at 4.80 V. This can be attributed to the 3D structure, which can strengthen this capture ability [25].

The application of lithium metal anode is a Gordian technique to ensure the capacity density of SLBs. In order to investigate the effect of 3D SiO_2_ NFs on the interfacial stability between SPE and lithium anode, we evaluated Li | SPE | Li batteries with a current of 0.1 mA cm^−2^ at 60 °C. The Li | PLS/SiO_2_ particles | Li exhibited an obvious polarization and short circuit at 120 h as shown in Figure 4. The poor cycling performance of Li | PLS/SiO_2_ particles | Li cells were related to the uneven dispersion of inorganic fillers. The aggregation area of PLS/SiO_2_ particles is a mechanical weakness and a concentrated area for lithium-ion plating/stripping. Lithium dendrites thus easily form at this weak point. In the case of Li | PLS/3D SiO_2_ NFs | Li, stable cycling upon 1000 h with a small overpotential was found. Cross-linked 3D SiO_2_ NFs, which can promote the formation of a uniform electric field and uniform mechanical stress distribution, plays a role in suppressing the growth of lithium dendrites [42,43]. Therefore, SPE with 3D SiO_2_ NFs can effectively suppress the growth of dendritic lithium. 

To investigate the feasibility of 3D SiO_2_ NF framework in SSBs, cells with LFP cathodes/Li anodes were assembled, and their electrochemical performance was investigated. As can be seen in Figure 5a,b, Li | PLS/3D SiO_2_ NFs | LFP exhibited the typical potential plateaus of 3.40 V for discharging and 3.46 V for charging at 0.1 C. With a coulombic efficiency of 95.7%, its initial specific capacity was 167.9 mAh g^−1^, and it only suffered 3.28% capacity degradation and exhibited little polarization after 100 cycles. Moreover, Li | PLS/3D SiO_2_ NFs | LFP maintained a superior coulombic efficiency of about 99.7% during the cycle. However, Li | PLS/SiO_2_ particles | LFP only delivered an initial specific capacity of 135 mAh g^−1^ with a coulombic efficiency of 92.2%, and it suffered rapid capacity fading and increasing polarization within 20 cycles. The enhanced battery performance can be attributed to the low ASR and good compatibility with electrode materials of PLS/3D SiO_2_ NFs. The rate performance of Li | PLS/3D SiO_2_ NFs |LFP batteries are shown in Figure 5c,d. The ohmic polarization of Li | PLS/3D SiO_2_ NFs | LFP increased with increasing current. The specific discharge capacity of Li | PLS/3D SiO_2_ NFs | LFP decreased from 166.9 mAh g^−1^ at 0.1 C to 168.3 and 159.8 mAh g^−1^ at 0.2 and 0.5 C, respectively. However, the coulombic efficiency of Li | PLS/3D SiO_2_ NFs | LFP maintained above 99.7% at elevated rates, and the specific discharge capacity could be quickly recovered to 170.4 mAh g^−1^ when the current density returned to 0.1 C, revealing the excellent rate performance of Li | PLS/3D SiO_2_ NFs | LFP. 

Electrolytes should be mechanically stable inside the cells to prevent physical contact of electrodes, even when the cells are operating in an extreme environment at high temperatures [44]. To investigate the effect of 3D SiO_2_ NFs on the thermal stability of electrolytes, PLS/3D SiO_2_ NFs and PLS/SiO_2_ particles were heated directly with an alcohol lamp, and the results are shown in Figure 6. The electrolyte quickly turned black and gradually recovered to white color after 2 min due to the gasification and thermal decomposition of PLS. The 3D SiO_2_ NFs framework remained complete and suffered no effect after heating, as shown in Figure 6a and Appendix A. Under the same condition, the structure of the PLS/SiO_2_ particles was destroyed, leaving only the separated SiO_2_ particles and thus losing the ability to separate the electrodes.

In order to further demonstrate the effect of 3D SiO_2_ NFs on the thermal stability of electrolytes, AC impedances of PLS/SiO_2_ particles and PLS/3D SiO_2_ NFs from 60 to 500 °C with a ramping rate of 10 °C min^−1^ were measured and are shown in Figure 7a. The resistance of PLS/SiO_2_ particles decreased rapidly at approximately 140 °C due to the thermal shrinkage, whereas the PLS/3D SiO_2_ NFs showed typical AC impedance curves and a stable resistance value below 300 °C (Figure 7b,c). The resistance increased rapidly above 400 °C, indicating a decomposition of PLS. To verify the hypothesis, the TGA curves of PEO, PLS/SiO_2_ particles, and PLS/3D SiO_2_ NFs were examined. As displayed in Figure 7d, there was an obvious weight drop for PLS/SiO_2_ particles and PLS/3D SiO_2_ NFs at 150–200 °C, which was in response to the volatilization of SCN in an open environment. The starting decomposition temperature of PEO was 340 °C in the N_2_ atmosphere, which was consistent with the results of the increased resistance of PLS/3D SiO_2_ NFs, as shown in Figure 7c. Based on the above results, PLS/3D SiO_2_ NFs can cut off the ion transport channels by decomposing PLS while separating cathodes and anodes through the solid framework of 3D SiO_2_ NFs. This is similar but superior to the shutdown function of the polypropylene/polyethylene/polypropylene (PP/PE/PP) separator, as a much wider safe temperature range is realized in PLS/3D SiO_2_ NFs.

The OCVs of Li | PLS/SiO_2_ particles | LFP and Li | PLS/3D SiO_2_ NFs | LFP pouch cells as a function of temperature were carried out to analyze the effect of 3D SiO_2_ NFs on the thermal stability of batteries. As shown in Figure 7e, OCV of the Li | PLS/SiO_2_ particles | LFP rapidly dropped to 0 V at approximately 135 °C. Clearly, the drop of OCV in Li | PLS/SiO_2_ particles | LFP could be imputed to the internal short circuit of punch cell due to the thermal shrinkage of PLS/SiO_2_ particles. However, the OCV of Li | PLS/3D SiO_2_ NFs | LFP remained stable up to 200 °C. To further verify the effect of 3D SiO_2_ NFs on the thermal stability of battery performance, the treated Li | PLS/3D SiO_2_ NFs | LFP was recycled at 60 °C. As shown in Figure 7f, the charging and discharging behavior of Li | PLS/3D SiO_2_ NFs | LFP was completely unaffected, which further emphasizes the excellent thermal stability of PLS/3D SiO_2_ NFs. 

The above improvement of battery performance with the aid of PLS/3D SiO_2_ NFs can be explained by Figure 8. As shown in Figure 8a,b, the shrinkage or decomposition of PLS will destroy the structure of the PLS and PLS/SiO_2_ particles, which will lead to the contact of electrodes, further causing the short circuit of batteries. However, the ceramic framework of PLS/3D SiO_2_ NFs can prevent the contact of anodes and cathodes when PLS is decomposed, thus ensuring the safety of batteries. Besides, with the aid of SiO_2_ particulate fillers, the gathering of ceramic fillers leads to uneven mechanical stress and electric field distribution of PLS/SiO_2_ particles. The aggregated areas of PLS/SiO_2_ particles are the mechanical weak points as well as concentrated areas for lithium-ion plating/stripping. Therefore, lithium dendrites can easily form and pierce those weak points (Figure 8b). Meanwhile the uniform ceramic framework of the PLS/3D SiO_2_ NFs is beneficial to the even transport of lithium ions, which can effectively inhibit the growth of lithium dendrites (Figure 8c).

## 4. Conclusions

In summary, a high-safety SPE with a framework of 3D SiO_2_ NFs and PLS hosts was prepared via a simple casting method. Taking advantage of flexible 3D SiO_2_ NFs, PLS/3D SiO_2_ NFs with a tensile strength of 7.4 MPa could be made with thickness as small as 27 μm. In combination with the high conductivity of approximately 9.32 × 10^−5^ S cm^−1^ at 30 °C, the PLS/3D SiO_2_ NFs exhibited an area specific resistance of 29.0 Ω cm^2^, which is favorable for the improved energy density of batteries. In addition, this cross-linked framework of 3D SiO_2_ NFs offered a uniform dispersion of highly conductive interfaces between PLS and SiO_2_, resulting in stable cycling over 1000 h at 0.1 mA cm^−2^ and 60 °C of the Li | PLS/3D SiO_2_ NFs | Li cell. Benefiting from the low ASR and good compatibility with electrode materials of PLS/3D SiO_2_ NFs, the Li | PLS/3D SiO_2_ NFs | LFP batteries showed excellent cycling performances at 60 °C. The initial specific capacity at 0.1 C was 167.9 mAh·g^−1^, and 96.7% of the discharge capacity could be retained after 100 cycles. Moreover, the framework of 3D SiO_2_ NFs separated the electrodes when the PLS was thermally decomposed above 400 °C, and cells automatically shut down in consequence. Therefore, the solid lithium batteries based on PLS/3D SiO_2_ NFs reported here offer high safety at elevated temperatures above 400 °C. The design of PLS/3D SiO_2_ NFs is expected to provide an instructive strategy to achieve thermal safety of SPEs for SLBs.

## Figures and Tables

**Figure 1 polymers-12-01324-f001:**
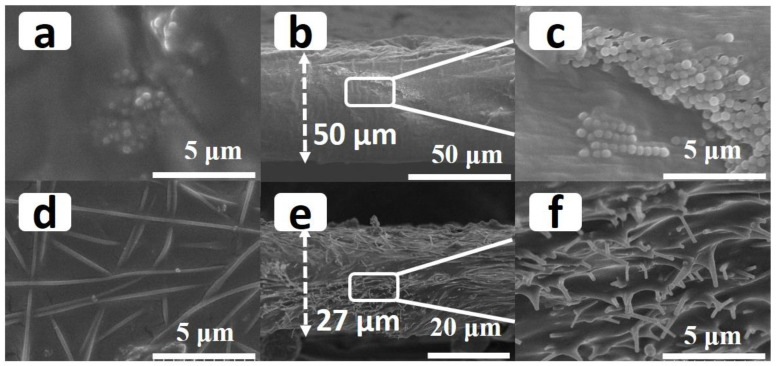
The top view (**a**) and cross-sectional views (**b**,**c**) of polyethylene oxide–lithium (bis trifluoromethyl) sulfate–succinonitrile (PLS)/SiO_2_ particles. Agglomeration of SiO_2_ nanoparticles in the PLS can be seen. The top view (**d**) and cross-sectional views (**e**,**f**) of PLS/3D SiO_2_ nanofibers (NFs)_._ Uniform distribution of PLS in the framework of 3D SiO_2_ NFs can be seen.

**Figure 2 polymers-12-01324-f002:**
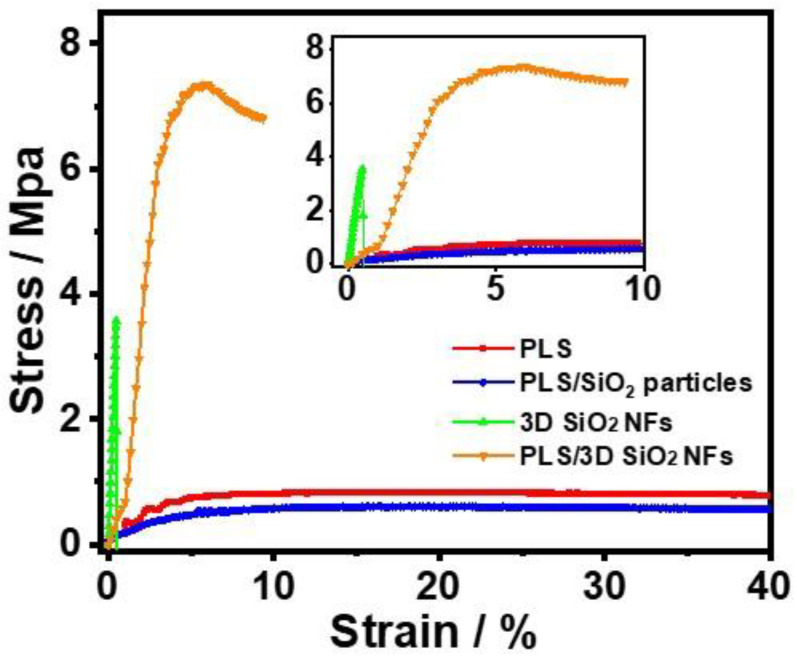
Stress–strain curves of the 3D SiO_2_ NFs and as-prepared solid polymer electrolytes (SPEs).

**Figure 3 polymers-12-01324-f003:**
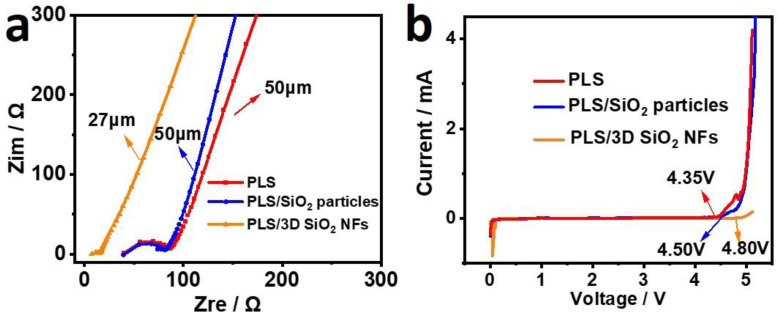
(**a**) AC impedance curves of PLS, PLS/SiO_2_ particles, and PLS/3D SiO_2_ NFs. (**b**) Linear sweep voltammetry (LSV) profiles of PLS, PLS/SiO_2_ particles, and PLS/3D SiO_2_ NFs with a range of 0–6 V at 0.5 mV s^−1^.

**Figure 4 polymers-12-01324-f004:**
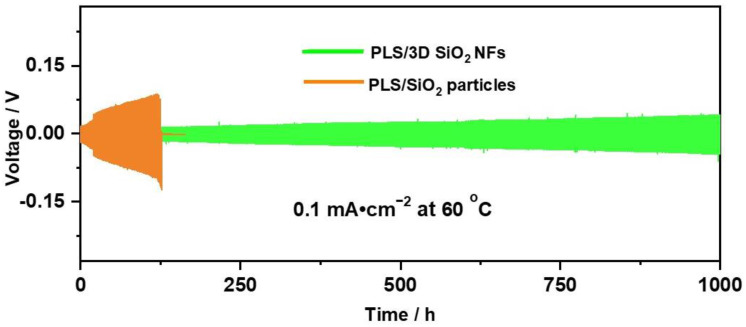
The voltage profiles of Li symmetrical cells with PLS/SiO_2_ particles and PLS/3D SiO_2_ NFs at a current density of 0.1 mA cm^−2^ with 0.5 h stripping/0.5 h plating alternating steps.

**Figure 5 polymers-12-01324-f005:**
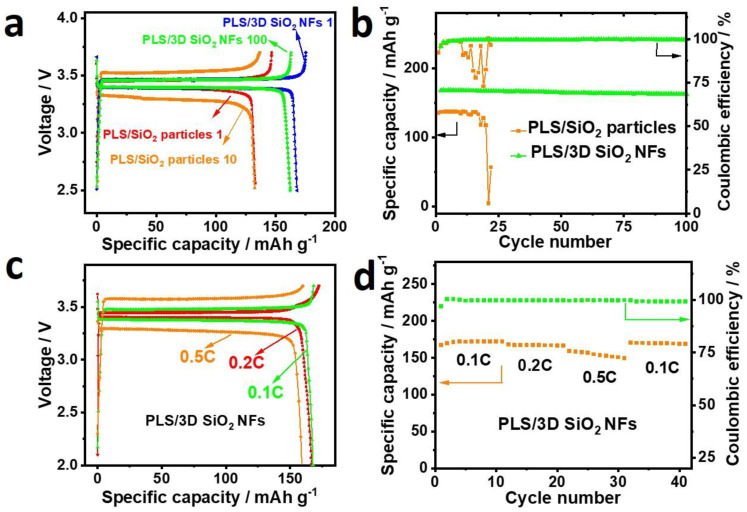
Charge and discharge curves (**a**) and cycling performance (**b**) of Li | PLS/SiO_2_ particles | LiFePO_4_ (LFP) and Li | PLS/3D SiO_2_ NFs | LFP. Charge and discharge curves (**c**) and rate performance (**d**) of Li | PLS/3D SiO_2_ NFs | LFP.

**Figure 6 polymers-12-01324-f006:**
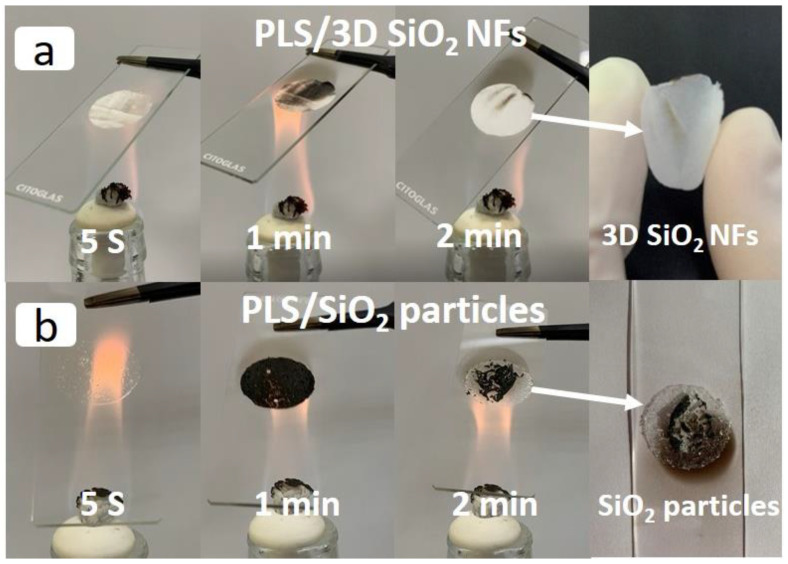
Digital images of PLS/3D SiO_2_ NFs (**a**) and PLS/SiO_2_ particles (**b**) heated with an alcohol lamp.

**Figure 7 polymers-12-01324-f007:**
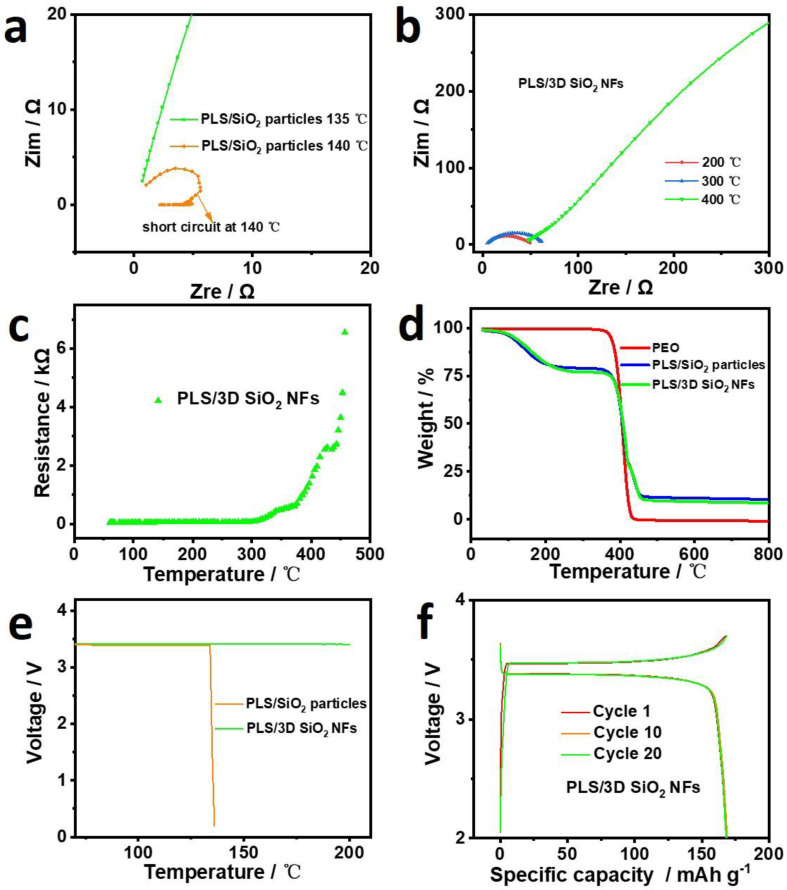
(**a**) AC impedance curves of PLS/SiO_2_ particles at 135 and 140 °C. AC impedance curves (**b**) and resistance (**c**) of PLS/3D SiO_2_ NFs at 200–400 °C. (**d**) TGA profiles of PEO, PLS/SiO_2_ particles, and PLS/3D SiO_2_ NFs with a heating rate of 10 °C min^−1^. (**e**) Open circuit voltage (OCV) of Li | PLS/SiO_2_ particles | LFP and Li | PLS/3D SiO_2_ NFs | LFP pouch cells as a function of temperature at 60–200 °C. (**f**) Charge and discharge curves of Li | PLS/3D SiO_2_ NFs | LFP pouch cells after OCV treatment.

**Figure 8 polymers-12-01324-f008:**
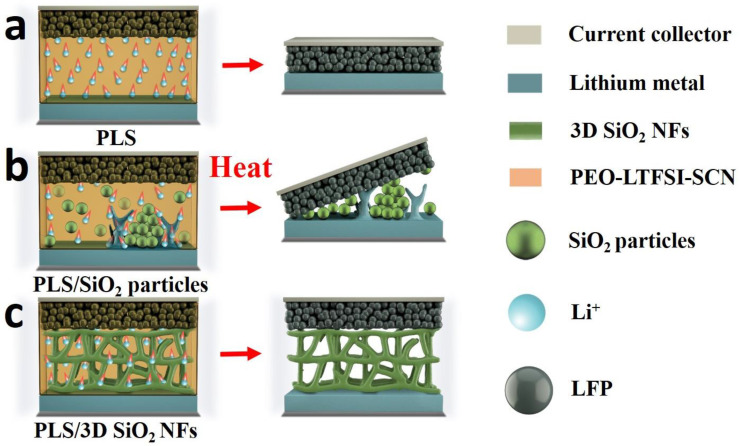
Schematic diagram of cells with (**a**) PLS, (**b**) PLS/SiO_2_ particles, and (**c**) PLS/3D SiO_2_ NFs.

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
