# Peer review of "Solid Polymer Electrolytes with Flexible Framework of SiO2 Nanofibers for Highly Safe Solid Lithium Batteries"

_polymers, 2020, doi:10.3390/polym12061324_

Round 1
Reviewer 1 Report
Summary and General Comment:
This work entitled “Solid polymer electrolytes with a flexible framework of SiO2 nano-fibers for highly safe solid lithium batteries” reports a polyethylene oxide-lithium (bis trifluoromethyl) sulfate-succinonitrile (PLS) and a framework of three-dimensional SiO2 nano-fibers (3D SiO2 NFs) are prepared as a composite electrolytes consisting of polymers and three-dimensional fillers. The total conductivity of the electrolyte approaches 9.32×10-5 S cm-1. The cell assembled with the composite electrolyte shows a discharge capacity of 167.9 mAh g-1 and a cycle life of 100 cycles. A Minor Revision suggestion is provided to the authors for addressing the mentioned concerns.
Additional Comments:
- What is the method used to improve the contact between the electrode and the electrolyte?
- What is the interface impedance between electrolyte and electrode?
- Why the Coulombic efficiency increases initially? Why the Coulombic efficiency increases initially from a low value?
- The mass loading of LFP should be reported. The mass of the electrolyte should be reported.
- What would be the energy density of the cell applied the reported solid polymer electrolyte?
- In the stress–strain curves, the curves of the 3D SiO2 NFs and PLS / 3D SiO2 NFs are hard to read. An inset enlarged the curves is suggested.
Author Response
Dear Editor,
Thanks a lot for reviewing our manuscript and giving us an opportunity to revise our manuscript, we appreciate editors and professional reviewers very much for their positive and constructive comments and suggestions.
We have uploaded the revised manuscript with changes marked-up using red highlights, so that the revisions can easily be identified. And below is the point-to-point responses to the comments:
The manuscript has been resubmitted to your journal. We are looking forward to your positive response.
Sincerely Yours,
Xiangxin Guo
Response to reviewer 1:
Thank you very much for your careful reviewing and checking:
- What is the method used to improve the contact between the electrode and the electrolyte?
Reply: The well contact between the electrode and the electrolyte is critical to have good performance for solid state batteries. In this work, we designed organic-inorganic composite electrolyte, with such flexible electrolyte a tight interface with the electrode can be achieved. The interface is further optimized under 60 oC for 10 h to reduce the interfacial resistance.
- What is the interface impedance between electrolyte and electrode?
Reply: The interface impedance of Li | PLS/3D SiO2 NFs | LFP is less than 600 Ω under 30 oC before hot treatment, it can be less than 300 Ω after treatment.
- Why the Coulombic efficiency increases initially? Why the Coulombic efficiency increases initially from a low value?
Reply: The Coulombic efficiency increases initially is due to the irreversible side reaction between electrodes and electrolyte. After several cycles, the Coulombic efficiency is stable at 99.7% which is a very excellent performance.
- The mass loading of LFP should be reported. The mass of the electrolyte should be reported.
Reply: The loading of LFP is 1.0 mg/cm2. The mass of electrolyte is 4.0 mg/cm2. It has been added on page 4 and line 17 and line 13, the revised places are marked in red in the revised manuscript.
- What would be the energy density of the cell applied the reported solid polymer electrolyte?
Reply: The pouch cell made in this work just contained one piece of electrode, so the energy density of the cell cannot be calculated. The theoretical energy density of the cell applied to the reported solid polymer electrolyte can be more than 250 wh/kg with LFP and 350 wh/kg with NCM.
- In the stress–strain curves, the curves of the 3D SiO2 NFs and PLS / 3D SiO2 NFs are hard to read. An inset enlarged the curves is suggested.
Reply: Thanks for your valuable suggestion. We have replaced the Fig. 2 with new ones and an inset enlarged the curves is supported.

Reviewer 2 Report
Dear authors, I have enjoyed reading this work. It is interesting and clearly presented. The strategy is appealing and characterization is sound.
I have one comment, regarding the effect on electrochemical stability of the SiO2 particles or NF. I do not quite understand the reason why the addition of silica particles can decrease the moisture, since these particles usually carry moisture themselves (also they are acidic). Likewise for the NFs.
Is the stability increase related to the size of the interface between the polymer electrolyte and the silica (particles or fibers?) Is the interaction with the polymer electrolyte stabilising the polymer's labile sites? Wouldn't it be worthwhile to try to understand this point better?
Author Response
Dear Editor,
Thanks a lot for reviewing our manuscript and giving us an opportunity to revise our manuscript, we appreciate editors and professional reviewers very much for their positive and constructive comments and suggestions.
We have uploaded the revised manuscript with changes marked-up using red highlights, so that the revisions can easily be identified. And below is the point-to-point responses to the comments:
The manuscript has been resubmitted to your journal. We are looking forward to your positive response.
Sincerely Yours,
Xiangxin Guo
Response to review 2:
- I have one comment, regarding the effect on electrochemical stability of the SiO2 particles or NF. I do not quite understand the reason why the addition of silica particles can decrease the moisture, since these particles usually carry moisture themselves (also they are acidic). Likewise for the NFs.
Reply: The SiO2 nano-fibers were sintering under 600 oC for 2h, the moisture water in SiO2 can be removed completely. SiO2 can absorb moisture water when mixed with the organic electrode due to its excellent affinity with water.
- Is the stability increase related to the size of the interface between the polymer electrolyte and the silica (particles or fibers?) Is the interaction with the polymer electrolyte stabilising the polymer's labile sites? Wouldn't it be worthwhile to try to understand this point better?
Reply: Thank you for your valuable comment. The electrochemical stability of SPE increases with adding the organic fillers since the uniform interface is conducive to the distribution of the electric field. Compared with the SiO2 particles that agglomerated easily, the 3D nano-fibers have more uniform interface. It is very likely the size of the interface between the polymer electrolyte and the silica is related to the stability of the electrolyte. The study of the interface size on the stability of the polymer is very worthwhile for studying.